# Synthesis and characterisation of peroxypinic acids as proxies for highly oxygenated molecules (HOMs) in secondary organic aerosol

Sarah S. Steimer[1], Aurélie Delvaux[1], Steven J. Campbell[1], Peter J. Gallimore[1], Peter Grice[1], Duncan J. Howe[1], Dominik Pitton[2], Magda Claeys[3], Thorsten Hoffmann[2], Markus Kalberer[1]

[1]Department of Chemistry, University of Cambridge, Cambridge, CB2 1EW, UK
[2]Institute of Inorganic and Analytical Chemistry, University of Mainz, 55128 Mainz, Germany
[3]Department of Pharmaceutical Sciences, University of Antwerp, 2610 Antwerp, Belgium

*Correspondence to*: Sarah S. Steimer (ss2349@cam.ac.uk), Markus Kalberer (markus.kalberer@atm.ch.cam.ac.uk)

**Abstract.** Peroxy acids were recently found to be involved in new particle formation in the atmosphere, and could also substantially contribute towards particle toxicity. However, a lack of suitable analytical methods for the detection and characterisation of peroxy acids in the particle phase is currently hindering the quantitative investigation of their contribution to these important atmospheric processes. Further development of appropriate techniques and relevant standards is therefore urgently needed. In this study, we synthesised three peroxypinic acids, developed a liquid chromatography separation method and characterised them with tandem mass spectrometry. The observed fragmentation patterns clearly distinguish the different peroxypinic acids from both the acid and each other, showing several neutral losses previously already observed for other peroxy acids. Both monoperoxypinic acids were found to be present in secondary organic aerosol generated from ozonolysis of α-pinene in laboratory experiments. The yield of monoperoxypinic acid formation was not influenced by humidity. Monoperoxypinic acid quickly degrades on the filter, with about 60% lost within the first 5 hours. This fast degradation shows that time delays in traditional off-line analysis will likely lead to severe underestimates of peroxy compound concentrations in ambient particles.

## 1 Introduction

In recent years, organic peroxy compounds have emerged as important particle-phase constituents of secondary organic aerosol (SOA). They are discussed as components of a family of compounds denoted as highly oxygenated molecules (HOMs), important in new particle formation (Ehn et al., 2014; Rissanen et al., 2015), and they could be involved in particle toxicity due to their oxidising properties, contributing to overall particle-bound reactive oxygen species (ROS) concentrations (Arashiro et al., 2017; Jiang et al., 2017). Multiple studies have shown that exposure to atmospheric aerosols is correlated with a wide variety of negative health outcomes (Brunekreef and Holgate, 2002; Dockery et al., 1993; Hoek et al., 2013; Nel, 2005). While it is not yet known which particle constituents are the main cause of particle toxicity, ROS, which summarises various oxygen-containing species with strong oxidative capacities, are implicated as a potential main contributor (Dellinger et al., 2001; Li et al., 2003; Tao et al., 2003).

In addition, it was recently shown that HOMs, which are suggested to form through autoxidation in the gas phase, are highly relevant for the initial stages of SOA formation (Ehn et al., 2014; Rissanen et al., 2014). Many studies state that HOMs have O:C ratios of $\geq 0.7$ (Mentel et al., 2015; Mutzel et al., 2015). There is no generally accepted definition of HOMs, but they typically contain multiple hydroperoxy and/or peroxy acid groups (Mentel et al., 2015; Rissanen et al., 2015).

Therefore, the characterisation and quantification of peroxy compounds in the particle phase has become an important issue. However, there is currently a lack of suitable analytical methods, which is exacerbated by the lack of appropriate standards for method development and validation. One subgroup of peroxy compounds are peroxy acids, which, apart from their relevance for atmospheric chemistry, are widely used as chemical reagents, disinfectants and bleaching agents (Holah et al., 1990; Luukkonen and Pehkonen, 2017; Milne, 1998; Ramirez, 2008; Reinhardt and Borchers, 2009). Due to their

widespread practical applications, several methods have been developed both for the quantification of the overall peracid content (Davies and Deary, 1988; Greenspan and MacKellar, 1948), as well as separation and detection of specific peracids (Cairns et al., 1975; Effkemann et al., 1998; Kirk et al., 1992). These methods use relatively unspecific detectors such as flame ionisation, electrochemical and UV/VIS detectors. This poses a problem for the analysis of ambient particles, as they can contain several thousands of organic species, demonstrating the need for authentic standards. Mass spectrometry, in

particular coupled with chromatography, provides a method to characterise and identify specific compounds. Recently, several studies have utilised mass spectrometry to analyse different types of organic peroxy compounds with potential atmospheric relevance (Witkowski and Gierczak, 2013; Zhao et al., 2018; Zhou et al., 2018; Ziemann, 2003). However, to our best knowledge no such studies exist for peroxy acids. In a previous paper, we synthesised one cyclic and several linear aliphatic peroxy acids and showed their separation and detection with liquid chromatography-tandem mass spectrometry

(HPLC-MS/MS) (Steimer et al., 2017). This study revealed that all investigated peroxy acids show common and specific mass spectrometry fragmentation patterns, which could be used as diagnostics to identify unknown peroxy acids in complex organic mixtures. However, as all structures synthesised in Steimer et al. (2017) are likely not prominent in secondary organic aerosol, we expand in this study the characterisation of peroxy acids towards structures that are likely present in atmospheric SOA particles. Monoperoxypinic acids have been suggested as potential products of α-pinene and β-pinene

ozonolysis (Docherty et al., 2005) and have been tentatively identified in α-pinene and β-pinene SOA using online MS/MS (Reinnig et al., 2009; Zhou et al., 2018). They are suggested to form in the gas phase via $HO_2$ chemistry (Docherty et al., 2005). In this study, we have synthesised three peroxypinic acids, identified their structure via their MS/MS and NMR spectra and investigated their presence and stability in α-pinene SOA generated in laboratory experiments under varying conditions. The structural similarity of these peroxy acids with HOMs (present in a wide range of SOA particles) makes

them ideal and unique proxies and surrogate standards for future studies aiming to quantify the role of HOMs in organic aerosols.

## 2 Experimental section

### 2.1 Chemicals and Synthesis

Sulfuric acid ($H_2SO_4$, 95-98%), hydrogen peroxide ($H_2O_2$, 50%, stabilised), α-pinene ($C_{10}H_{16}$, ≥ 98%), cis-pinonic acid, bromine, dioxane, diethyl ether and hydrochloric acid were bought from Sigma-Aldrich. D(+)-Camphoric acid ($C_{10}H_{16}O_4$, ≥ 99%) was purchased from Acros Organics. Dichloromethane was bought from Carl-Roth, sodium sulfate was supplied by Acros, sodium hydroxide was purchased from Merck and charcoal pellets were provided by Fluka. Water, formic acid (0.1% in water) and acetonitrile, all in OPTIMA® LC/MS grade, were purchased from Fisher Scientific.

cis-Pinic acid was synthesised using a similar procedure as described by Moglioni et al. (2000). 2.8 g cis-pinonic acid was dissolved in 50 mL dioxane. 8.2 g sodium hydroxide was dissolved in 196 mL water and loaded with 7.5 g bromine (0 °C). The prepared sodium hydrobromite solution was added dropwise to the pinonic acid solution (30 min, −15 °C). After 2 h of stirring at 0 °C and overnight (approx. 16 h) at room temperature, the resulting solution was extracted three times each with 20 mL dichloromethane. 60 mL of a 40% sodium hydrogensulfate solution, prepared by dissolving sodium sulfate in water, was added to the water phase. The pH was set to 1 using hydrochloric acid. This solution was extracted 5 times using 5 x 40 mL diethyl ether. The collected ethereal phases were dried using 1 g sodium sulfate. After filtration and evaporation, the slightly yellow powder was purified by dissolving it in 2 mL diethyl ether and addition of 5 pellets of charcoal. After subsequent filtration followed by evaporation, the yield of cis-pinic acid was estimated to be 2.5 g (88%) with a purity of 96% regarding the educt. To confirm the identity of the synthesised compound $^1H$, $^{13}C$, DEPT, COSY, HSQC and HMBC NMR spectra were collected using residual $CHD_2CN$ as the internal standard. cis-Pinic acid: $^1H$ NMR ($CD_3CN$, 500 MHz) $δ_H$ 0.94 (s, 3H, H5), 1.20 (s, 3H, H6), 1.82 (m, 1H, H2$_α$), 2.03 (m, 1H, H2$_β$), 2.3 (m, 3H, H1,H8), 2.74 (dd, J = 10.3 Hz, J' = 7.9 Hz, 1H, H3). $^{13}C$ NMR ($CD_3CN$, 500 MHz) $δ_C$ 17.8 (C5), 25.2 (C2), 30.0 (C6), 35.2 (C8), 38.9 (C1), 42.9 (C4), 46.4 (C3), 174.2 (C7), 174.4 (C9). A full overview of all NMR spectra used for the assignment is given in the supplement (Fig. S1-S6)

A mixture of diperoxypinic acid ($C_9H_{14}O_6$) and two different monoperoxypinic acids ($C_9H_{14}O_5$) was synthesised from pinic acid ($C_9H_{14}O_4$) using a procedure adapted from Parker et al. (1957), who describe the synthesis of various aliphatic diperoxy acids. The structures of the synthesised compounds are shown in Fig. 1. For the synthesis, 10 mg of pinic acid was dissolved in 22 µL of concentrated sulfuric acid and the resulting solution was cooled to 10-15 °C in an ice water bath. Under constant stirring, 13 µL of 50% $H_2O_2$ was slowly added dropwise to the solution. After 180 min reaction time, 100 µL of ammonium sulfate solution (350 mg·ml$^{-1}$, 4 °C) was added to the product solution. Since no precipitate was formed, the solution was mixed with 0.5 ml 20:80 water:acetonitrile and stored in a freezer (−22 °C) until further analysis. The product solution consists of two phases, of which the upper one (acetonitrile phase) was used for analysis. To confirm the identity of the synthesised products, the monoperoxypinic acid and diperoxypinic acid fraction of the chromatographic separation were

collected four times each and the two resulting solutions were dried at 30 °C under a steady flow of nitrogen and then dissolved in deuterated acetonitrile for subsequent NMR measurements. The NMR spectra of the monoperoxypinic acid fraction confirm that both monoperoxypinic acid isomers are present. However, they are not stable under the present conditions and have largely already reacted back to pinic acid by the time of the first measurement. Additional spectra taken

several hours after the first one show a continuing decay into pinic acid. The NMR spectra of the diperoxypinic acid fraction are initially dominated by diperoxypinic acid, although both monoperoxypinic acid isomers and pinic acid are also present. Spectra taken several hours after the initial measurement show a marked increase in pinic acid and both monoperoxypinic acid isomers as diperoxypinic acid decays further. A more detailed discussion of the NMR results and corresponding figures (Fig. S7-S17) can be found in the supplement.

**2.2 Flow tube experiments**

An aerosol flow tube (2.5 L) was used to produce α-pinene SOA. An overview of the setup for sampling under humid conditions is shown in Fig. 2. Ozone ($O_3$) was produced by irradiating a flow of synthetic air (200 mL/min) with a UV-lamp (185/254 nm, Appleton Woods®). The resulting average $O_3$ concentrations were 21-22 ppm under dry conditions and 16-19 ppm under humid conditions as some of the $O_3$ was removed in the Gore-Tex tube of the humidifier (Fig. 2). Gaseous α-

pinene was introduced into the flow tube by passing $N_2$ (200 mL/min) over 500 µL of liquid α-pinene (about 340 ppm initial concentration, measured by PTR-MS according to the procedure described in Giorio et al. (2017)), which results in a residence time of approximately 6.3 min. Under these conditions, the reaction is limited by $O_3$, which according to model calculations is consumed within ~20 s under both humid and dry conditions. The lifetime of $O_3$ and α-pinene in the flow tube was estimated using the AtChem (http://atchem.leeds.ac.uk) numerical box-model alongside the Master Chemical

Mechanism (MCM) v3.3.1 (http://mcm.leeds.ac.uk) (Jenkin et al., 1997; Saunders et al., 2003). For the experiments at higher humidity (~85%), a humidifier was added to the setup. A scanning mobility particle sizer (SMPS) was used to monitor the concentration of produced SOA before and after sample collection. The SMPS consists of a TSI 3080 differential mobility analyser (DMA) connected to a TSI 3775 condensation particle counter (CPC). The average particle mass concentration in the flow tube was about $5 \times 10^4$ µg·m$^3$, assuming a density of 1 g·cm$^3$, with a mode of 200 nm for the number concentration.

Under these conditions, α-pinene partitioning to the particles is still negligible (<1%) and while wall partitioning could be significant, it is unlikely to directly modify the observed aerosol composition due to the lower volatility of the products. The produced α-pinene SOA was collected on Durapore® membrane filters (0.1 µm pore size, 47 mm diameter, Merck) for a sampling period of 45 min. Partitioning of α-pinene followed by condensed-phase reactions on the filter might occur and could change the aerosol composition compared to lower mass loadings, although the charcoal denuder should have removed

the majority of organic gases and $O_3$, making this less likely. After sample collection, 100 µL of a 0.60 mg/L camphoric acid solution (20:80 $H_2O$:ACN) was added to each filter as an internal standard. The filters were extracted by vortexing in 1.5 mL ACN for 5 min. The extracts were then evaporated down to a volume of 300 µL under a steady flow of nitrogen at 30°C to minimize the time between collection and analysis. While the increased temperature might lead to thermal decomposition,

this effect should be small and is likely outweighed by the reduction of the total analysis time. As a final step, 1.2 mL of 0.1% formic acid in water was added to better match initial mobile phase conditions of the HPLC analysis.

To investigate the influence of humidity on the formation of monoperoxypinic acid, a total of 13 filter samples were collected: 7 under dry conditions (≤3% RH), and 6 for α-pinene oxidation at ~85% RH. The average filter mass loading was 0.87 mg for the dry and 0.55 mg for the humid oxidation. All filters were extracted immediately after sampling. Three LC-MS/MS runs were conducted for each filter extract.

The degradation of peroxypinic acid in α-pinene SOA was investigated by subsequently collecting SOA on four filters under dry conditions. After collection of the final filter sample, the filters were cut into quarters and divided into four identical composite samples, containing one quarter from each filter. The first composite sample was extracted immediately, following the same procedure as previously described. The three additional composite samples were extracted after being stored in their filter boxes at room temperature and under protection from UV radiation for up to 70 h to simulate typical field sampling conditions. All extracts were analysed by HPLC-MS/MS multiple (2-5) times.

## 2.3 HPLC-MS/MS Analysis

All samples were analysed via HPLC-ESI-MS/MS. An Accela system (Thermo Scientific, San Jose, USA) equipped with a T3 Atlantis C18 column (3 µm; 3.0 × 150 mm; Waters, Milford, USA) was used for the chromatographic separation. The detector was a high-resolution LTQ Orbitrap Velos mass spectrometer (Thermo Scientific, Bremen, Germany) with a heated electrospray ionisation (HESI) source. All data were analysed with Thermo Xcalibur 2.2.

For analysis of the synthesised standard, 10 µL of the product mixture were diluted with 1 mL of an 80:20 water:acetonitrile mixture. For the flow tube experiments, filters were extracted according to the procedure described in the previous section and measured without further sample processing.

Pinic acid and its synthesised peroxy acids were separated using gradient elution at a flow rate of 200 µL/min. The mobile phase was composed of a mixture of 0.1% formic acid in water (solvent A) and acetonitrile (solvent B). The initial concentration of 20% B was kept constant for 2 min and then increased to 23% over the course of 20 min. Thereafter, it was further increased to 90% within the next 6 min. Over the next 5 min, it was then decreased back to 20% and subsequently kept at this concentration for an additional 10 min, resulting in a total analysis time of 43 min.

The mass spectrometer was calibrated using Pierce® ESI Negative Ion Calibration Solution (Thermo Scientific, Rockford, USA). All samples were analysed in negative ionization mode, using the following source parameters: spray voltage −3.3 kV, source heater temperature 250 °C, capillary temperature 275 °C, sheath gas flow 40 arbitrary units, auxiliary gas flow 10 arbitrary units, no sweep gas flow and S-lens RF level 70%. Mass spectra were collected in full scan mode over a mass range of $m/z$ 100−650, using the lock mass of the deprotonated dimer of formic acid at $m/z$ 91.0037 and a resolution of 100 000 at $m/z$ 400. For the majority of measurements, alternating MS/MS scans were performed using a global mass list including the masses of deprotonated pinic, monoperoxypinic and diperoxypinic acid as well as deprotonated camphoric acid, which was used as an internal standard in the flow tube experiments. The MS/MS scans were performed with a

resolution of 60 000, an isolation width of 4 mass units and a mass range that was $m/z$ 50−220 for pinic acid and its derivatives and 50−210 for camphoric acid. The collision energy level was 27% for pinic acid and its derivatives and 22% for camphoric acid. A few additional runs were performed with isolation width 1.5, leading to a strongly reduced signal of monoperoxypinic acid isomer I. For investigation of the monoperoxyacid dimeric adduct, alternating MS³ scans were

performed instead with the mass of the deprotonated monoperoxypinic acids selected for the second fragmentation. In this case, the collision energy level was 27% for both fragmentation steps, the resolution was 60 000, the mass range was $m/z$ 55−410 and the isolation width was 4 mass units for the first fragmentation and 2.5 mass units for the second one. In all cases, collision-induced dissociation (CID) was used to induce fragmentation.

## 2.4 Evaluation of MS/MS spectra

Elemental compositions were assigned with a mass tolerance of 6 ppm, including only the elements carbon, oxygen and hydrogen. The list of MS/MS fragments contains all $m/z$ values smaller than the precursor ion which have a relative abundance above the chosen threshold of >1%. The same procedure was repeated with a background subtracted version of the spectrum. The final list of product ions is based on the background subtracted mass list, but additionally excludes $m/z$ values which did not have an assigned elemental composition, which could be dismissed as product ions based on their

assigned elemental composition or whose extracted ion chromatogram showed a poor overlap with the base peak chromatogram of the MS/MS measurement. While product ions and their relative abundances were selected based on the background subtracted mass list, the masses themselves were taken from the raw data as they showed higher mass accuracy. Since the chromatographic peaks of the two monoperoxy acids overlap (Fig. 3), special care was taken to separate the two spectra. Only the third of the chromatographic peaks furthest removed from their neighbour was used for the evaluation. In

addition to the monomers, there were also dimeric adducts of monoperoxypinic acid detected at the same chromatographic peaks. Since they have the same retention time as the monomers, these compounds are ESI artefacts and not actually present in the sample. MS³ spectra were taken for the mass of the dimeric monoperoxypinic acid adduct ($m/z$ 403.16159), which is more prominent for isomer I. The resulting MS³ spectrum of the dimeric adduct from the isomer I peak shows the same product ions as the MS/MS spectrum of the monomer, but with lower abundances for those common to the two isomers . It

was therefore taken to be closer to the pure spectrum of isomer I and used to characterise MS/MS fragmentations of that compound. Similarly, initial MS/MS conditions lead to low isolation efficiency of the deprotonated molecule of isomer I so that the MS/MS spectrum for isomer II was not significantly influenced by isomer I.

## 2.5 Data analysis of flow tube measurements

To investigate the presence of peroxypinic acid in the samples, extracted ion chromatograms of the peracid main product

ions were selected. If a peak was found at the appropriate retention time, its MS/MS spectrum was compared to that of the corresponding standard. The integrated peak of the extracted ion chromatogram of $m/z$ 183.06 was used to determine the

relative yield of monoperoxypinic acid isomer II. It was normalised to the peak area of the main product ion of camphoric acid, *m/z* 155.10, and to the mass loading of the filter.

## 3 Results and discussion

### 3.1 Liquid chromatography separation of the synthesised standards

5 The chromatogram shows that pinic acid and its peroxy acid derivatives, synthesised as described above, elute from the HPLC column in order of decreasing polarity, with pinic acid eluting first, followed by the two monoperoxy pinic acids and finally diperoxypinic acid (Fig. 3). While the acid and diperoxy acid are clearly separated from the monoperoxy acids, the two monoperoxypinic acids strongly overlap due to similar polarities. Peaks were assigned based on deprotonated analyte ions ([M − H]⁻), as well as the occurrence of different product ions for the two monoperoxypinic acid isomers.

10 ### 3.2 Mass spectrometry of pinic acid and the synthesised standards

All four analytes were detected as deprotonated molecules, i.e. [M − H]⁻ ions (Fig. S18). In the case of pinic acid and the two monoperoxypinic isomers, the deprotonated molecule is dominant in the chromatographic peak. In contrast, the mass spectrum of diperoxypinic acid was dominated by [M – H]⁻ ions of pinic and monoperoxypinic acid, which were about an order of magnitude more abundant than that of diperoxypinic acid. As described in previous papers (Harman et al., 2006; 15 Steimer et al., 2017), this likely indicates electrochemical reduction of the analyte (i.e., diperoxypinic acid) in the ion source. The pinic acid [M – H]⁻ ion was also present in the monoperoxypinic acid spectra, although about an order of magnitude less abundant than that of monoperoxypinic acid, which follows the trend of less efficient electroreduction of monoperoxy vs. diperoxy acids reported in our previous paper (Steimer et al., 2017). Tandem mass spectrometry was performed on all four analyte [M – H]⁻ ions, the results of which are summarised in Table 1. A graphic presentation of the MS/MS data can be 20 found in the supplement (Fig. S19). The product ions observed for pinic acid are in line with results from previous studies (Glasius et al., 1999; Yasmeen et al., 2011), where loss of $CO_2$ was observed as the main fragmentation process.

All neutral losses observed for pinic acid also occurred for both of the monoperoxypinic acid isomers. However, as previously observed for aliphatic peroxy acids, there are additional loss processes present compared to the acid (Steimer et al., 2017), four of which are shared between the two isomers. Two of the resulting fragment ion types, [M − $CH_2O_2$]⁻ and [M 25 − $CH_4O_4$]⁻, were observed in our previuous study for linear monoperoxydicarboxylic acids, while [M − $C_2H_2O_4$]⁻ was also observed for the only measured cyclic monoperoxydicarboxylic acid (monoperoxycamphoric acid) and [M − $C_2H_2O_3$]⁻ was not detected for any of the peroxy acids investigated in our previous study (Steimer et al., 2017).

The MS/MS spectra of the two isomers can be easily distinguished. While the main difference lies in the relative abundances of the various product ions, the spectrum of isomer I also shows several product ions not present for isomer II. Using this 30 information, it was possible to determine that the electroreduction of diperoxypinic acid in the ion source predominantly leads to formation of isomer II. Based on the available literature (Szmigielski et al., 2006; Yasmeen et al., 2010, 2011), we

suggest potential fragmentation pathways for the two isomers (Fig. S20). This allows the tentative assignment of isomer I and II as the monoperoxy pinic acid isomer with a methyl peroxycarboxyl substituent and a peroxycarboxyl substituent, respectively.

The diperoxypinic acid spectrum is dominated by loss of $CHO_3$, a process which was also observed for monoperoxycamphoric acid and monoperoxypinic acid isomer I. Two other relatively abundant neutral losses are peracid-specific fragments: $H_2O_2$, which was previously also observed in linear diperoxy acids, and $C_2H_2O_5$, which gives a minor contribution to the spectrum of monoperoxypinic acid isomer I.

In summary, the three peroxy acids synthesised here showed unique trends in fragmentation patterns for mono- and diperoxy acids, respectively. Some of these fragmentations were also observed for the 15 peroxy acids we characterised earlier (Steimer et al., 2017). The consistent fragmentation patterns of peroxy acids suggests that they might be suitable to identify unknown peroxy acids in SOA, e.g. as HOMs or ROS markers.

## 3.3 Flow tube experiments

The formation of peroxypinic acids in SOA formed through ozonolysis of α-pinene was investigated using the flow tube set up detailed above. The MS/MS spectra of the filter extracts showed presence of both monoperoxypinic acid isomers (both suggested to form during α-pinene ozonolysis (Docherty et al., 2005; Reinnig et al., 2009)), while there was no evidence of diperoxypinic acid formation. While both monoperoxypinic acid isomers were formed during α-pinene oxidation, the peak of isomer I was often too small for reliable integration, so that only isomer II was chosen for the following quantitative analyses. Even though isomer II could be reliably detected, it is only a minor product of the α-pinene oxidation, with about 1/5500 the peak area of pinic acid.

## 3.3.1 Humid vs. dry conditions

The relative yields of monoperoxypinic acid under dry and humid ozonolysis conditions were investigated experimentally and *via* the AtChem box model (https://atchem.leeds.ac.uk). The complete reaction scheme for the degradation of α-pinene was extracted from the MCM v3.3.1 (Jenkin et al., 1997; Saunders et al., 2003) via the MCM website (http://mcm.leeds.ac.uk/MCM). Gas-phase only simulations were performed for dark ozonolysis with $[\alpha\text{-pinene}]_0 = 300$ ppm and $[O_3]_0 = 20$ ppm. It is known that the gas-phase formation of peroxy acids can proceed via $HO_2$ chemistry, without direct involvement of $H_2O$ (Docherty et al., 2005; Eddingsaas et al., 2012). The calculated yield of monoperoxypinic acid per $O_3$ molecule was $\sim 6 \times 10^{-5}$ ($\sim 1.2$ ppb) under dry conditions and was insensitive to RH (0-100%) and initial precursor concentrations (1-300 ppm). This confirms the unimportant role for water vapour in the gas-phase formation of monoperoxypinic acid not only as a reactant, but also in terms of indirect effects on e.g. the concentrations of precursor species such as Criegee intermediates. The yield of monoperoxypinic acid was ~1/500 that of pinic acid in the simulations, compared to the experimental estimate above (~1/5500).

In the condensed phase, however, the formation of peracids from the carboxylic acid is reversible, with presence of liquid water shifting the equilibrium towards the acid (d'Ans and Frey, 1912; Parker et al., 1955). Increased humidity, leading to an increased fraction of water in the particle phase, could therefore lead to less peroxypinic acid in α-pinene ozonolysis SOA. We proceeded with experiments to investigate whether humidity-dependent gas-particle partitioning and/or condensed-phase reactions, not accounted for in the model, could influence peroxy acid yields.

The relative yield of monoperoxypinic acid isomer II in SOA was compared under dry (≤3% RH) and humid (~85% RH) conditions. The results of the comparison are shown in Fig. 4. Within the current limits of uncertainty, no difference in monoperoxypinic acid production was found for the two reaction conditions. This indicates that hydrolysis of monoperoxypinic acid is not a significant loss process under the studied conditions (i.e. reactant concentrations and reaction time). Available studies of the hydrolysis of peracetic acid show that its hydrolysis kinetics strongly depend on the reaction conditions, such as temperature and pH (Dul'neva and Moskvin, 2005; Yuan et al., 1997), leading to large variations in decomposition rates. The fact that the peroxypinic acid yield per SOA mass does not depend on humidity agrees with observations made by Docherty et al., (2005), who found no dependence of the organic peroxide yield per SOA mass on humidity. Previous studies of the humidity dependence of individual peroxy compounds were focused on small molecules predominantly residing in the gas phase (e.g. Hasson et al., 2001; Huang et al., 2013) and are therefore not directly comparable with our results. However, the fact that different correlations with humidity were found for different peroxy compounds demonstrates the need for investigation of individual compounds. The main factors limiting precision of our measurement are low signal intensity, uncertainty of the measurement of filter mass loading and peroxypinic acid degradation on the filter, as described in the following section.

### 3.3.2 Peroxy acid degradation over time

We investigated the stability of monoperoxypinic acid isomer II in SOA over time when the filter was stored at room temperature after collection. In many studies which characterise SOA composition in detail at a molecular level, filter samples, especially from field campaigns, are collected many hours to days or weeks before analysis. While these samples are usually kept at low temperatures for long-term storage, they are collected at room temperature, which can take longer than 24 h and there is often significant delay between sampling and storage. For thermally unstable compounds such as peroxides and peroxy acids, this might result in decomposition prior to analysis and thus risks underestimating the abundance of such compounds.

Monitoring of the amount of peroxy acid on the collected filters over almost three days clearly shows that this compound significantly degrades (Fig. 5). About 60% of monoperoxypinic acid is lost within the first five hours. Given that the measured samples were composites of four subsequently collected filters, the average age of the SOA sample before extraction is ca. 100 min. Therefore, a significant amount of monoperoxypinic acid will already have been lost at the time of analysis. Repeated measurements of the extracts over 22 hours showed that there is also a change in concentration over time in the liquid phase, however significantly less pronounced with a maximum of a factor two difference to the initial

measurement. We therefore attempted to measure all extracts as soon as possible after extraction, using the same number of repeats. This was not always possible for the experiment shown as purple diamonds in Fig. 5, contributing to the uncertainty of the results.

## 4 Conclusions

In this study, we successfully synthesised three peroxypinic acids and showed that they can be distinguished from each other and the analogous carboxylic acid via their retention times in HPLC-MS and their specific MS/MS spectra. This technique can therefore be used to identify peroxypinic acids in SOA samples. We have shown that one of the peroxypinic acids, monoperoxypinic acid isomer II, was present in laboratory-generated α-pinene SOA. There was no observed effect of humidity on the production of monoperoxypinic acid from α-pinene, i.e. the reactions times in our flow tube set up were possibly too short to observe any potential decay due to hydrolysis. It was shown that monoperoxypinic acid quickly degrades, with about 60% lost within the first 5 hours. This demonstrates that filters need to be analysed as soon as possible after collection to avoid serious underestimation for such compounds, which presents a major problem particularly during field campaigns, where such immediate analysis is often not feasible. Ideally online techniques would need to be developed to quantify such unstable compounds in atmospheric aerosols.

## Acknowledgement

Funding: S.S.S. acknowledges support from the Swiss National Science Foundation (project no. 162258). Funding by the European Research Council (ERC starting grant 279405) and the European Union's Horizon 2020 research and innovation programme through the EUROCHAMP-2020 Infrastructure Activity under grant agreement No 730997 is acknowledged.

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

**Table 1: MS/MS fragmentations of [M − H]⁻ ions from pinic acid and its peroxyacid analogues.**

| Compound | Deprotonated molecule [M − H]⁻ | m/z (Δ ppm) | Product ion m/z (rel. abund.)[a] | Neutral loss(es) |
|---|---|---|---|---|
| Pinic acid | $C_9H_{13}O_4^-$ | 185.08193 (-0.002) | 141.09194 (100) | $CO_2$ |
| | | | 167.07127 (16) | $H_2O$ |
| | | | 123.08155 (4) | $CH_2O_3$ |
| Monoperoxypinic acid isomer I | $C_9H_{13}O_5^-$ | 201.07706 (1.060) | 111.08149 (100) | $C_2H_2O_4$ |
| | | | 155.07127 (38) | $CH_2O_2$ |
| | | | 139.07639 (37) | $CH_2O_3$ |
| | | | 140.08423 (10) | $CHO_3$ |
| | | | 157.08694 (8) | $CO_2$ |
| | | | 183.06619 (8) | $H_2O$ |
| | | | 127.07643 (5) | $C_2H_2O_3$ |
| | | | 84.02166 (3) | $C_5H_9O_3$ |
| | | | 115.00367 (3) | $C_5H_{10}O$ |
| | | | 129.05568 (3) | $C_3H_4O_2$ |
| | | | 99.04513 (2) | $C_4H_6O_3$ |
| | | | 111.04514 (2) | $C_3H_6O_3$ |
| | | | 121.06583 (2) | $CH_4O_4$ |
| | | | 184.07401 (1) | $OH$ |
| | | | 95.08662 (1) | $C_2H_2O_5$ |
| Monoperoxypinic acid isomer II | $C_9H_{13}O_5^-$ | 201.07711 (1.293) | 183.06605 (100) | $H_2O$ |
| | | | 157.08678 (35) | $CO_2$ |
| | | | 139.07625 (31) | $CH_2O_3$ |
| | | | 155.07116 (10) | $CH_2O_2$ |
| | | | 111.08142 (5) | $C_2H_2O_4$ |
| | | | 127.07635 (2) | $C_2H_2O_3$ |
| | | | 121.06577 (2) | $CH_4O_4$ |
| Diperoxypinic acid | $C_9H_{13}O_6^-$ | 217.07232 (2.573) | 156.07910 (100) | $CHO_3$ |
| | | | 155.07132 (16) | $CH_2O_3$ |
| | | | 111.08152 (13) | $C_2H_2O_5$ |

[a] Average of three MS/MS spectra, with exception of monoperoxypinic acid isomer I (average of two MS³ spectra, see section 2.4 for discussion)

| | | | 183.06628 (7) | $H_2O_2$ |
| | | | 171.06625 (3) | $CH_2O_2$ |
| | | | 173.08189 (2) | $CO_2$ |
| | | | 199.06122 (1) | $H_2O$ |
| | | | 74.00093 (1) | $C_7H_{11}O_3$ |
| | | | 127.07646 (1) | $C_2H_2O_4$ |

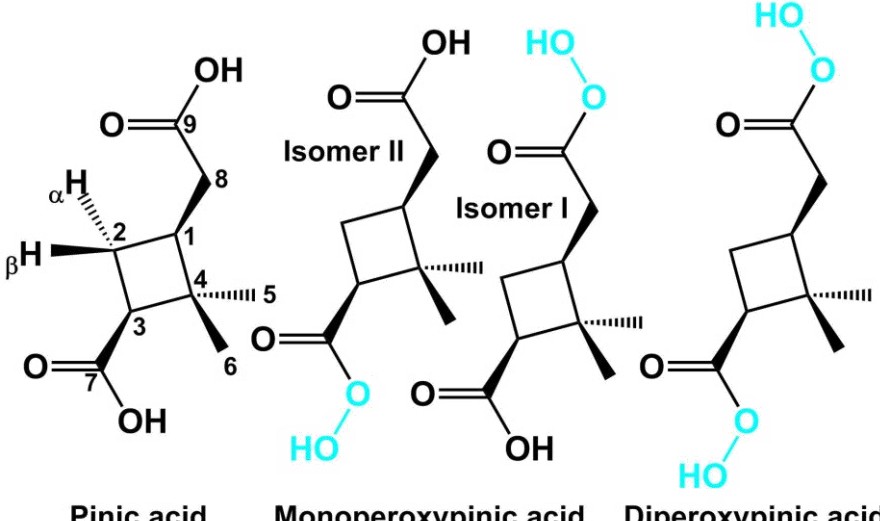

**Figure 1 Structures of *cis*-pinic acid and its peroxy acid derivatives synthesised and characterised in this study.**

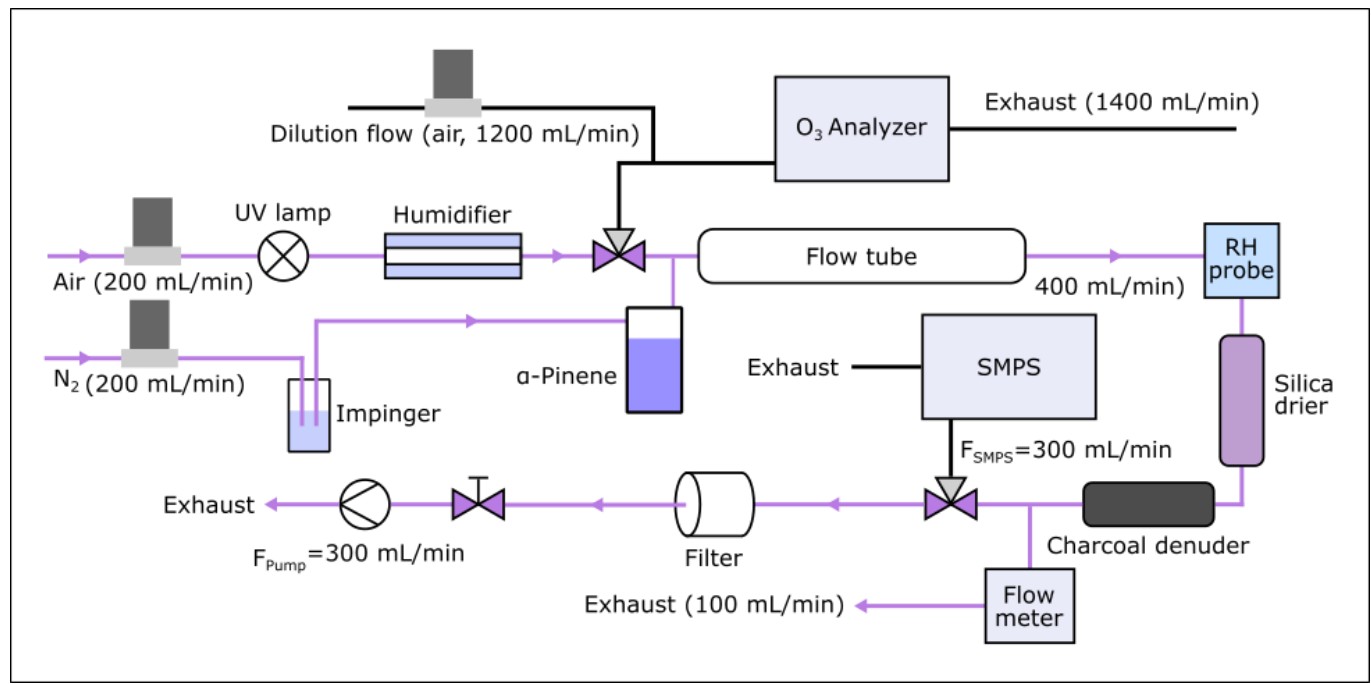

**Figure 2: Flow tube setup for oxidation under humid conditions. The pink lines show the gas and aerosol flow path during sample collection, when O₃ analyser and SMPS are not connected.**

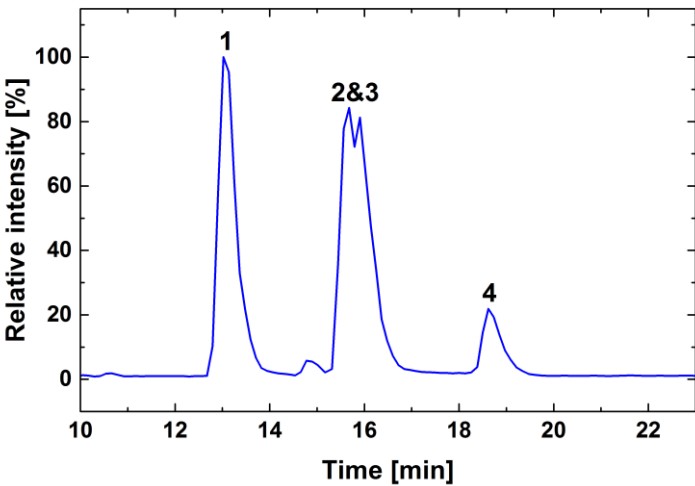

5     **Figure 3: Base peak chromatogram of the synthesised mixture for mass range $m/z$ 100−650, showing the separation of pinic acid (1), monoperoxypinic acid isomers I and II (2, 3) and diperoxypinic acid (4).**

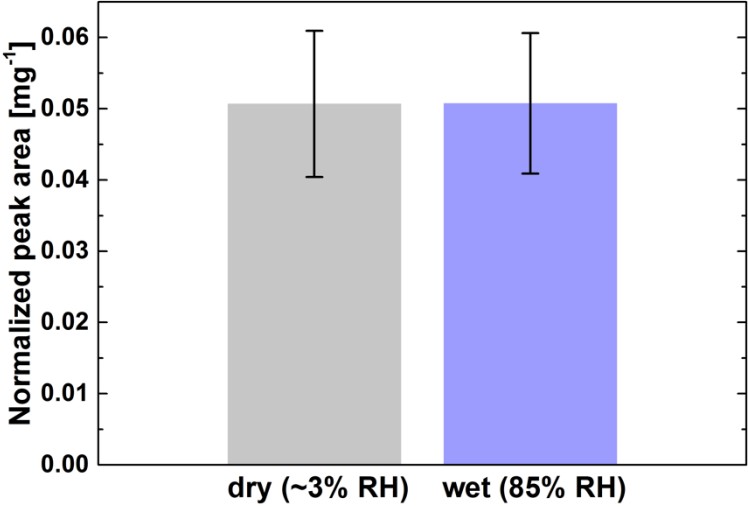

**Figure 4: Monoperoxypinic acid isomer II as a product of α-pinene oxidation under dry and humid conditions; error bars show the standard deviation between repeat measurements. The peak area was normalised to the peak area of the internal standard and the aerosol mass on the filter.**

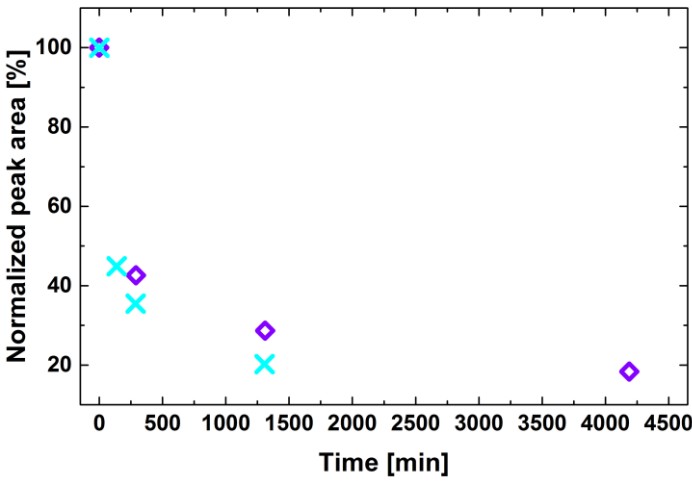

**Figure 5 : Degradation of monoperoxypinic acid isomer II on the filter illustrating its short lifetime of only a few hours. Diamonds and crosses represent two different repeats of the degradation experiment.**