# Peer review of "Synthesis and characterisation of peroxypinic acids as proxies for highly oxygenated molecules (HOMs) in secondary organic aerosol"

_Atmospheric Chemistry and Physics, 2017_

## Referee Comment (RC1) · Anonymous Referee #1 · 23 Jan 2018

This manuscript describes the synthesis of two monoperoxypinic acid isomers (designated I and II) along with the diperoxyacid in a one-pot procedure. The peroxyacids are characterized by HPLC/ESI-MS, including MS2 analysis, but pure compounds were not isolated. Dimer species observed at the retention times of the monoperoxy acids were shown to be artifacts generated in the ESI source. SOA was generated from alpha pinene and analysis demonstrated the presence of the monoperoxy isomers, with isomer II predominating. The effect of RH on SOA formation was investigated and found to be insignificant. Stability of the monoperoxy isomers in SOA collected on filters was

also investigated and rapid decomposition was observed, indicating that even rapid work-up will likely result in significant underestimation of peroxyacid concentration.

This manuscript is important as an illustration of the need for synthesis of authentic standards for verification and quantitation of SOA components. Furthermore, the flow tube experiments support the growing realization that peroxyacids, among other highly oxidized species may be significant contributors to SOA and its biological effects. The manuscript is clearly written and work logically developed. Eventual publication is therefore strongly recommended. However, the report has several shortcomings that must be addressed prior to publication.

1. In general, synthetic standards should be isolated and characterized as pure compounds. While the mass spectrometric analysis presented in this manuscript is convincing that the structures proposed for the synthetic targets are correct, additional physicochemical data, for example, 1H NMR, is desirable for definitive proof. The authors report that the monoperoxy acids could not be well resolved, but effort directed towards achieving resolution is not clear. For example, a promising possibility would be the use of a HILIC column, such as recently reported for separation of carboxylic acids in SOA (J. Chromatogr. A 2011, 1218, 4417– 4425). The column used in this citation was also a 3 mm column, which would be adequate for collection of sufficient sample for NMR analysis. If the authors did investigate the separation more thoroughly, this information should be included.

2. Why was the LC/ESI-MS analysis not performed using a UPLC column? A UPLC version of the same column would likely provide significant improvement in resolution. It would be surprising if the authors did not have access to UPLC. Mass spectrometric data on better-resolved peaks could have acquired even though resolution on the semi-preparative scale column was difficult to achieve.

3. The observation of different MS2 spectra for isomers I and II indicates that 1-electron reduction in the ESI source did not make an important contribution to the mass spectra of the monoperoxy acids and that the anions of the more acidic carboxylic acids were the parent species. The fragmentation patterns provide sufficient information to distinguish between the two proposed structures, and the authors should be able to tentatively assign structures to isomers I and II. Although the literature on EIS-MS of carboxylic acids is sparse, there is sufficient precedent to propose plausible pathways to the observed product ions which are unique to each structure.

4. Graphic presentation of the actual MS and the MS2 data would be helpful to the readers. The mass spectra could be presented as supporting information.

5. The investigation of the effects of RH rules out the formation of the peroxyacids via a hydrolytic pathway, but does not preclude an oxidative pathway in the condensed phase. In the gas phase, peroxyacids can form via HO2 chemistry (e.g., Atmos. Chem. Phys. 2012, 12, 6489–6504) without the participation of H2O. Therefore insensitivity of yield to RH is not surprising, and the discussion might be amended to reflect this. The order of magnitude discrepancy between the observed and predicted relative yields of monoperoxypinic acids is probably not explicable entirely by peroxyacid degradation during work-up, and illustrates the importance of quantitation using authentic standards to improve models.

An experiment that would be informative with regard to whether the peroxyacids form in the gas or condensed phases would be to monitor the gas phase species by CIMS if the authors have access to such instrumentation. Although interesting, this experiment would not be a prerequisite to publication since access to CIMS instrumentation is required.

---

## Referee Comment (RC2) · Anonymous Referee #2 · 5 Feb 2018

The manuscript entitled "Synthesis and characterization of peroxypinic acids as proxies for highly oxygenated molecules (HOMs) in secondary organic aerosol" investigates the effect of RH on the formation of peroxycarboxylic acids. For this study important peracids were synthesized. In particular, the degradation of peroxides on filter samples is described that should be considered for all future work. The content is very good and the manuscript well written but lacks of many important points (e.g. literature comparison, chromatograms and spectra as evidence for the described findings etc.). As the described findings highly affect the aerosol community I recommend publication, but only after the following points were carefully addressed. I would also recommend to change the title. The reasons for this are explained in detail within the review.

General comments:

Introduction: To my opinion the introduction is too much concentrated on highly oxidized molecules (HOMs). Even that this group of compounds is a so-called "hot topic" the compounds that are described in the manuscript do not belong to HOMs. In the literature it is stated that HOMs contain hydroperoxide functions. The compounds described in the present manuscript are traditional peracids. Furthermore, HOMs are highly oxidized (O:C $\geq$ 1) and contain usually carbonyl groups. They are formed by autoxidation in the gas phase. The formation of peracids might proceed mainly via the reaction of an acylperoxy radical with $HO_2$ (Niki et al., 1985) and their O:C ratio is too small. Thus a basic discussion of formation pathways yielding peracids is largely missing in the present manuscript. Therefore it is questionable to me why peroxypinic acid is a proxy for HOMs. Thus the title is very misleading. Maybe the title should be reconsidered and also the relation between HOMs and peracids should not be highlighted or at least it should be discussed in a better and more logical way.

In the work by Ehn et al., or Riissanen et al., it is not stated the HOMs contain peracid structures. A discussion of the possible formation mechanism would be also helpful to understand the effect of RH that is one major focus of this manuscript.

Besides this discrepancy the introduction lacks of several references and thus, gives a very superficial impression. Several methods exist quantifying at least organic peroxides as a sum parameter (e.g., Docherty et al., 2005, Mutzel et al., 2013). Also studies are published characterising organic peroxides in SOA by LC/MS analysis (Krapf et al., 2016,

Zhao et al., 2018, Ziemann et al., 2003). In particular the work by Zhao et al., should be recognized within the present manuscript as they also synthesised peroxy compounds and characterised them with LC/MS.

Page 3, line 10: Pinic acid was only characterised by 1H-NMR. Please provide the NMR spectrum in the manuscript and calculate the purity of pinic acid. According to the NMR characterisation given in the manuscript an assignment of the signals to the H-atoms in the molecule is complicated. Please number the carbon atoms in Figure 1 and use these numbers to clearly assign the H-atoms to the signal recorded in the NMR. Furthermore, I assume that "complex adsorption" means multiplett? Please use the exact NMR wording for the interpretation.

The NMR characterisation of the peroxy compounds is completely missing. Please add this to the manuscript together with the corresponding NMR spectra. Please indicate the level of purity. How can the authors ensure the identity and purity of the synthesized compounds without NMR characterisation? In particular, this is very important for the peroxy compounds as they are synthesised for the first time. And also the purity will highly effect the quantification results.

Is there a reason that the characterisation is only done with 1H-NMR? The amount of material is enough to prepare 13C, NOESY or COSY spectra to provide a complete characterisation of all synthesised compounds. Also the corresponding MS spectra of all synthesised compound should be shown.

Page 3, line 20: The experimental conditions seem to be very high. What is the reason the run the experiments under those high conditions? How was the concentration of a-pinene measured? This chosen concentration contradict also the relation to HOMs. HOMs formation and in particular, the contribution of HOMs to the early particle growth becomes more important under low mass loadings. Under higher mass loadings the contribution of HOMs decrease as the contribution of subsequent chemistry of first-generation oxidation products (semi-volatile oxidation products such as pinonaldehyde etc.) starts to increase.

Page 4, line1: I´m wondering that the samples very dried at 30°C,. Why not at room temperature? Can the authors ensure that none of the peroxides decompose under this

temperature? Was the influence of the temperature investigated? How was the volume of 300 uL measured?

Page 5, line 18: It is not mentioned which efforts were made to separate the two overlapping peaks. Please show chromatogram and describe which parameters were tested to improve the separation. In addition, it is very questionable to use unseparated peaks for quantification. Even that it is stated that the second peak seems to be small Page 7, line 14), a reliable quantification should be done with well-separated peaks. Additionally an important picture is missing illustrating the BPC of the standard, the BPC from the flow tube experiment under humid conditions and the BPC under dry conditions.

Page 7, line 19: It is somehow not logical to describe the synthesised peracids as HOMs and to predict their formation with a model that does not contain any HOMs formation. As it is stated above. HOMs are formed via autoxidation during the initial phase of oxidation. Therefore MCM to predict and understand their formation is not suitable. The formation of peracids in MCM follows the traditional radical chemistry. This can be applied to the present peracids but it is not applicable to HOMs. Therefore a clear separation of both topics should be done! Furthermore, if the present peracids would be HOMs a discussion about their formation in the condensed phase would not make any sense (Page 7, line 26) as HOMs supposed to be formed solely in the gas phase. I also miss the corresponding pictures illustrating the simulation with AtChem vs. the experiment data.

Page 7, line 22: The yield should be also given as fraction in SOA. This is very common in studies investigating the contribution of peroxides to SOA formation. This would also enable a comparison to other literature studies which is also missing in the manuscript. Based on these values a better discussion of the effect of RH can be done.

Page 7, line 31: A comparison to literature studies is largely missing, e.g. Huang et al., 2013 investigated the influence of RH on hydroperoxides. The authors should also include other potential mechanism/precursor that can be affected by RH, like the Criegee intermediate.

Page 8, line 15: It is stated that peroxopinic acid degrades over time. Which other products were observed? I would expect the formation of decomposition products like pinic acid.

How were the samples treated between the repetitive analysis? Were they stored in a fridge or at room temperature? Were they always protected against UV light?

Minor comments

Page 1, Line 21: change particle phase to particle-phase

Page 2, Line 26: According to IUPAC nomenclature "Sulphuric acid" should be changed to "sulfuric acid". This is also the case in Page 3, line 17.

---

## Referee Comment (RC3) · Anonymous Referee #3 · 12 Feb 2018

General Comments

In this manuscript the authors describe results of an experimental study in which they synthesized 3 peroxyacid compounds expected to be formed in the ozonolyisis of a-pinene, characterized their mass spectra, and then used this information with liquid chromatography to search for them in secondary organic aerosol formed from this re-action. One of the peroxyacids was detected in the SOA and its yield quantified. The authors also show that peroxyacids decompose in SOA on timescales of hours, and so are difficult to measure in long filter samples. The method development and evalux

ation, and SOA experiment were well done and demonstrate that this approach could be used for analyzing these compounds and other peroxyacids in SOA systems. This approach represents an advance in the area of molecular analysis of SOA, especially for organic peroxides, which are of growing importance because of new understanding regarding autoxidation chemistry. The manuscript is well written, and I recommend it be published after the following minor comments are addressed.

Specific Comments

1. Page 3, lines 23–25: The concentrations of a-pinene and ozone used in these experiments were extremely high. Is it possible that a-pinene partitions to particles, walls, or the filter and that some of the reaction occurs there? Some discussion of the differences between reactions conducted under these conditions and at more typical atmospheric concentrations seems warranted.

2. Page 3, line 26: For these reactant concentrations the ozone should be gone in a few seconds, so the statement that the reaction time is 6.25 min could be clarified. This may give the time for particle-phase reactions, but these will then continue after collection on the filter.

3. Because of the unstable nature of peroxides it seems that some of the conditions in the HPLC-MS/MS analysis could impact the analysis. For example, the use of 0.1% formic acid, and heater and capillary temperatures of 250 C and 275 C. Please comment on this.

4. Did the authors consider measuring the total peroxide content of their SOA so that they could estimate the fraction of total peroxides that their molecular analysis detects?

5. Might it be possible to collect particles in a cooled filter apparatus in order to reduce the decomposition of peroxides?

Technical Comments

None

---

## Author Comment (AC1) · 31 May 2018

We would like to thank the reviewers for their constructive feedback on the manuscript. Our answers to the different remarks are detailed below with comments from the reviewers in black, our answers in blue and suggested changes to the manuscript in italics.

**Reply to anonymous referee #1**

1. In general, synthetic standards should be isolated and characterized as pure compounds. While the mass spectrometric analysis presented in this manuscript is convincing that the structures proposed for the synthetic targets are correct, additional physicochemical data, for example, 1H NMR, is desirable for definitive proof.
To confirm the identity of the synthesized compounds, we collected the monoperacid and diperacid fraction of the chromatographic separation and subsequently performed NMR analysis. The compounds proved to be not sufficiently stable in the selected solvent to obtain pure NMR spectra of the different compounds. We do however feel confident that the measured 1-NMR and HSQC data of the collected fractions nevertheless strongly support the assigned structures. A detailed discussion of the NMR results can now be found in the supplement (Fig.S7-S17).

The authors report that the monoperoxy acids could not be well resolved, but effort directed towards achieving resolution is not clear. For example, a promising possibility would be the use of a HILIC column, such as recently reported for separation of carboxylic acids in SOA (J. Chromatogr. A 2011, 1218, 4417– 4425). The column used in this citation was also a 3 mm column, which would be adequate for collection of sufficient sample for NMR analysis. If the authors did investigate the separation more thoroughly, this information should be included.
We thank the reviewer for this interesting suggestion. Unfortunately, we have no HILIC column available for the current study but will consider this alternative separation technique in future studies. We decided to use the same solvents (acetonitrile and 0.1% formic acid in water) as in our previous study for easier comparison of the fragmentation patterns. We did however try different gradients with these two solvents, without any notable improvement in the separation of the two isomers.

2. Why was the LC/ESI-MS analysis not performed using a UPLC column? A UPLC version of the same column would likely provide significant improvement in resolution. It would be surprising if the authors did not have access to UPLC. Mass spectrometric data on better-resolved peaks could have acquired even though resolution on the semipreparative scale column was difficult to achieve.
We do not have access to UPLC instrumentation.

3. The observation of different MS2 spectra for isomers I and II indicates that 1-electron reduction in the ESI source did not make an important contribution to the mass spectra of the monoperoxy acids and that the anions of the more acidic carboxylic acids were the parent species. The fragmentation patterns provide sufficient information to distinguish between the two proposed structures, and the authors should be able to tentatively assign structures to isomers I and II. Although the literature on ESI-MS of carboxylic acids is sparse, there is sufficient precedent to propose plausible pathways to the observed product ions which are unique to each structure.
We have added an overview of potential fragmentation pathways to the Supplement. The proposed fragmentation schemes enable us to tentatively assign structures to isomer I and II. The following section has therefore been *added to the text (p.7, line 31):*
*"Based on the available literature* (Szmigielski et al., 2006; Yasmeen et al., 2010, 2011)*, we suggest potential fragmentation pathways for the two isomers (Fig. S20). This allows the tentative assignment of isomer I and II as the monoperoxy pinic acid isomer with a methyl peroxycarboxyl substituent and a peroxycarboxyl substituent, respectively.*

4. Graphic presentation of the actual MS and the MS2 data would be helpful to the readers. The mass spectra could be presented as supporting information.

*A supplement is now available which contains the requested MS and MS/MS spectra (Fig. S18 & S19). Appropriate cross references were added to the text of section 3.2.*

5. The investigation of the effects of RH rules out the formation of the peroxyacids via a hydrolytic pathway, but does not preclude an oxidative pathway in the condensed phase. In the gas phase, peroxyacids can form via $HO_2$ chemistry (e.g., Atmos. Chem. Phys. 2012, 12, 6489–6504) without the participation of $H_2O$. Therefore insensitivity of yield to RH is not surprising, and the discussion might be amended to reflect this. The order of magnitude discrepancy between the observed and predicted relative yields of monoperoxypinic acids is probably not explicable entirely by peroxyacid degradation during work-up, and illustrates the importance of quantitation using authentic standards to improve models.

*We agree that peroxy acids can form in the gas phase via $HO_2$ chemistry, without water vapour as a reactant, and have emphasised this in the revised manuscript. While a lack of RH dependence is therefore intuitively not surprising, the detailed MCM modelling confirms this quantitatively. Furthermore, the model results allow us to conclude that potential indirect effects of water vapour (e.g. changing the fate of precursor species such as Criegee intermediates) are also unimportant in determining the final yield of peroxypinic acid. We have modified the manuscript as follows (p.8):*

*"It is known that the gas-phase formation of peroxy acids can proceed via $HO_2$ chemistry, without direct involvement of $H_2O$ (Docherty et al., 2005; Eddingsaas et al., 2012). The calculated yield of monoperoxypinic acid per $O_3$ molecule was ~6 × $10^{-5}$ (~ 1.2 ppb) under dry conditions and was insensitive to RH (0-100%) and initial precursor concentrations (1-300 ppm). This confirms the unimportant role for water vapour in the gas-phase formation of monoperoxypinic acid not only as a reactant, but also in terms of indirect effects on e.g. the concentrations of precursor species such as Criegee intermediates."*

An experiment that would be informative with regard to whether the peroxyacids form in the gas or condensed phases would be to monitor the gas phase species by CIMS if the authors have access to such instrumentation. Although interesting, this experiment would not be a prerequisite to publication since access to CIMS instrumentation is required.

*We do not have access to a CIMS.*

**Reply to anonymous referee #2**

General comments:
Introduction: To my opinion the introduction is too much concentrated on highly oxidized molecules (HOMs). Even that this group of compounds is a so-called "hot topic" the compounds that are described in the manuscript do not belong to HOMs. In the literature it is stated that HOMs contain hydroperoxide functions. The compounds described in the present manuscript are traditional peracids. Furthermore, HOMs are highly oxidized (O:C ≥ 1) and contain usually carbonyl groups. They are formed by autoxidation in the gas phase. The formation of peracids might proceed mainly via the reaction of an acylperoxy radical with HO2 (Niki et al., 1985) and their O:C ratio is too small. Thus a basic discussion of formation pathways yielding peracids is largely missing in the present manuscript. Therefore it is questionable to me why peroxypinic acid is a proxy for HOMs. Thus the title is very misleading. Maybe the title should be reconsidered and also the relation between HOMs and peracids should not be highlighted or at least it should be discussed in a better and more logical way. In the work by Ehn et al., or Riissanen et al., it is not stated the HOMs contain peracid structures. A

discussion of the possible formation mechanism would be also helpful to understand the effect of RH that is one major focus of this manuscript.

We would like to note that, although some structures proposed and summarised in the recent literature as HOMs have hydroperoxy groups only, there is a significant number of proposed structures that contain both hydroperoxy **and** peroxy acid groups or peroxy acid groups only (e.g. Mentel et al., 2015; Rissanen et al., 2015). The text has been amended to clarify that either functional group can be present in HOMs (p.2).
*"Many studies state that HOMs have O:C ratios of ≥ 0.7 (Mentel et al., 2015; Mutzel et al., 2015). There is no generally accepted definition of HOMs, but they typically contain multiple hydroperoxy and/or peroxy acid groups* (Mentel et al., 2015; Rissanen et al., 2015)*."*

There is no unified definition of O:C in HOMs in the literature but many papers argue that O:C ≥ 0.7 should be used (Mentel et al., 2015; Mutzel et al., 2015) rather than 1 as suggested by the reviewer. The structures we investigate here are have O:C of 0.55 and 0.66 and thus have O:C very close to the definition of HOMs. This is now explicitly mentioned, see comment above.

We like to emphasise that a main aspect of this study was to provide the atmospheric community with a simple procedure to synthesise and characterise a realistic HOMs ***proxy*** for quantitative studies on HOMs as stated in the abstract. Therefore, we like to keep the current title. This is now more clearly clarified (p.2, line 29).
*"The structural similarity of these peroxy acids with HOMs (present in a wide range of SOA particles) makes them ideal and unique proxies and surrogate standards for future studies aiming to quantify the role of HOMs in organic aerosols."*

We are aware that peroxypinic acids are not formed via autoxidation and have stated this more clearly in the manuscript (p.2, line 26).

Besides this discrepancy the introduction lacks of several references and thus, gives a very superficial impression. Several methods exist quantifying at least organic peroxides as a sum parameter (e.g., Docherty et al., 2005, Mutzel et al., 2013). Also studies are published characterising organic peroxides in SOA by LC/MS analysis (Krapf et al., 2016, Zhao et al., 2018, Ziemann et al., 2003). In particular the work by Zhao et al., should be recognized within the present manuscript as they also synthesised peroxy compounds and characterised them with LC/MS.

We would like to focus the introduction on peroxy acids and not widen it to a general review of analytical technique to characterise peroxides. We have therefore added only a few references describing MS studies in which other potentially atmospherically relevant peroxy compounds were synthesized and studied. The study by Krapf et al. 2016 was not included here, as they did not synthesize any specific peroxy compounds.
We added Zhou et al. as additional reference for the tentative identification of monoperoxypinic acid in α-pinene aerosol.

This part of the introduction now reads (p.2, line 14): *"Mass spectrometry, in particular coupled with chromatography, provides a method to characterize and identify specific compounds. Recently, several studies have utilised mass spectrometry to analyse different types of organic peroxy compounds with potential atmospheric relevance (Witkowski and Gierczak, 2013; Zhao et al., 2018; Zhou et al., 2018; Ziemann, 2003). However, to our best knowledge no such studies exist for peroxy acids."*

Page 3, line 10: Pinic acid was only characterised by 1H-NMR. Please provide the NMR spectrum in the manuscript and calculate the purity of pinic acid. According to the NMR characterisation given in the manuscript an assignment of the signals to the H-atoms in the molecule is complicated. Please number the carbon atoms in Figure 1 and use these numbers to clearly assign the H-atoms to the signal recorded in the NMR. Furthermore, I assume that "complex adsorption" means multiplett? Please use the exact NMR wording for the interpretation.

We have measured additional C-NMR and 2-D NMR data and added the NMR spectra to the supplement. The description of the NMR spectra is now improved (p.3, line 18) and assignments of the carbon atoms and the hydrogen atoms are now given (some uncertainty remains for chemically different hydrogen atoms attached to the same carbon atom); the atom numbering has been added to figure 1. The purity in regards to the educt (*cis*-pinonic acid) is 96%, which has been added to the text (p.3, line 17). It was determined by integrating the peak of the methyl group of *cis*-pinonic acid at 0.87 ppm and the equivalent methyl group of *cis*-pinic acid at 1.02 ppm in the 1H-NMR and calculating the ratio of the two integrals. Spectra of both product and educt (in CDCl$_3$) that were used for this calculation are shown below.

[Figure]

*The NMR section now reads as follows (p. 3):" After subsequent filtration followed by evaporation, the yield of cis-pinic acid was estimated to be 2.5 g (88%) with a purity of 96% relative to the educt. To confirm the identity of the synthesised compound $^{1}H$, $^{13}C$, DEPT, COSY, HSQC and HMBC NMR spectra were collected using residual $CHD_2CN$ as the internal standard. cis-Pinic acid: $^{1}H$ NMR ($CD_3CN$, 500 MHz) $\delta_H$ 0.94 (s, 3H, H5), 1.20 (s, 3H, H6), 1.82 (m, 1H, H2$_\alpha$), 2.03 (m, 1H, H2$_\beta$), 2.3 (m, 3H, H1,H8), 2.74 (dd, J = 10.3 Hz, J' = 7.9 Hz, 1H, H3). $^{13}C$ NMR ($CD_3CN$, 500 MHz) $\delta_C$ 17.8 (C5 or C6), 25.2 (C2), 30.0 (C5 or C6), 35.2 (C8), 38.9 (C1), 42.9 (C4), 46.4 (C3), 174.2 (C7), 174.4 (C9). A full overview of all NMR spectra used for the assignment is given in the supplement (Fig. S1-S6)."*

The NMR characterisation of the peroxy compounds is completely missing. Please add this to the manuscript together with the corresponding NMR spectra. Please indicate the level of purity. How can the authors ensure the identity and purity of the synthesized compounds without NMR characterisation? In particular, this is very important for the peroxy compounds as they are synthesised for the first time. And also the purity will highly effect the quantification results.

Is there a reason that the characterisation is only done with 1H-NMR? The amount of material is enough to prepare 13C, NOESY or COSY spectra to provide a complete characterisation of all synthesised compounds.

*We would like to note that the synthesized compounds are present as a mixture and we did not further purify the compounds, which is why no attempts at absolute quantification are made in this study.*

*As discussed above, to confirm the identity of the synthesized compounds, we have now collected the monoperacid and diperacid fraction of the chromatographic separation and subsequently performed NMR analysis. The compounds proved to be not sufficiently stable in the selected solvent to obtain pure NMR spectra of the different compounds. We do however feel confident that the measured 1-NMR and HSQC data of the collected fractions nevertheless strongly support the assigned structures. A discussion of those NMR results can now be found in the text (p.4, line 2) and the supplement (Fig. S7-S17).*

Also the corresponding MS spectra of all synthesised compound should be shown.
*Mass spectra of the educt and all synthesized compounds have been added to the supplement (Fig. S18).*

Page 3, line 20: The experimental conditions seem to be very high. What is the reason the run the experiments under those high conditions? How was the concentration of α-pinene measured? This chosen concentration contradict also the relation to HOMs. HOMs formation and in particular, the contribution of HOMs to the early particle growth becomes more important under low mass loadings. Under higher mass loadings the contribution of HOMs decrease as the contribution of subsequent chemistry of first-generation oxidation products (semi-volatile oxidation products such as pinonaldehydeetc.) starts to increase.

*The reason to use high SOA precursor concentrations in this proof-of-concept study is the relatively low yield of monoperoxypinic acid; using concentrations closer to atmospheric conditions would have resulted in monoperoxypinic acid concentrations below the detection limit of our method.*
*α-Pinene was measured by PTR-MS according to the procedure described in Giorio et al. (2017); this is now shortly described on p.4, line 16.*

*We would like to emphasize again that we are aware that the formation of the synthesized peroxypinic acids does not proceed via autoxidation and their choice as a proxies for HOMs is purely due to chemical similarities, not similar formation processes. We emphasise this in the title and various part of the paper. The investigation of the peroxypinic acid formation process is independent*

from their suitability as HOMs proxies. We hope that this point is more clearly communicated in the current version of the manuscript.

Page 4, line1: I´m wondering that the samples very dried at 30°C. Why not at room temperature? Can the authors ensure that none of the peroxides decompose under this temperature? Was the influence of the temperature investigated? How was the volume of 300 uL measured?
The samples were dried at 30 °C to speed up the time between collection and analysis. This is now mentioned in the text on p.4, line 32 where we also acknowledge that we cannot rule out some decomposition of peroxides, although we believe that this should be a minor effect as we are heating the SOA extract only slightly above room temperature. The 300 μL were determined volumetrically.

Page 5, line 18: It is not mentioned which efforts were made to separate the two overlapping peaks. Please show chromatogram and describe which parameters were tested to improve the separation.
A chromatogram is shown in Fig. 3 of the original manuscript.  We decided to use the same solvents (acetonitrile and 0.1% formic acid in water) as in our previous study for easier comparison of the fragmentation patterns. To improve the separation, we tested multiple different gradients with these two solvents, varying total run time, starting concentration of the organic phase and the steepness of the gradient without any notable improvement in the separation of the two isomers.

In addition, it is very questionable to use unseparated peaks for quantification. Even that it is stated that the second peak seems to be small (Page 7, line 14), a reliable quantification should be done with well-separated peaks. Additionally an important picture is missing illustrating the BPC of the standard, the BPC from the flow tube experiment under humid conditions and the BPC under dry conditions.

As stated above, we were not able to obtain a better separation of the two monoperoxy acid peaks. Due to the small peak areas seen in the SOA samples, the peaks are reasonably separated, although some uncertainty remains.
We assume that BPC stands for base peak chromatogram. The base peak chromatogram of the synthesized standard mixture is shown in Fig.3 of the original manuscript. We have stated this now more clearly in the respective figure caption. We do not feel that the base peak chromatograms of the flow tube experiments would add any value to the manuscript since the peroxypinic acid only gives a comparatively small signal and is therefore not directly visible in the base peak chromatograms of the SOA. As stated in the data analysis section, quantification was done using the extracted ion chromatogram of the MS/MS measurement.

Page 7, line 19: It is somehow not logical to describe the synthesised peracids as HOMs and to predict their formation with a model that does not contain any HOMs formation. As it is stated above. HOMs are formed via autoxidation during the initial phase of oxidation. Therefore MCM to predict and understand their formation is not suitable. The formation of peracids in MCM follows the traditional radical chemistry. This can be applied to the present peracids but it is not applicable to HOMs. Therefore a clear separation of both topics should be done! Furthermore, if the present peracids would be HOMs a discussion about their formation in the condensed phase would not make any sense (Page 7, line 26) as HOMs supposed to be formed solely in the gas phase. I also miss the corresponding pictures illustrating the simulation with AtChem vs. the experiment data.
As stated previously, we are not suggesting that the investigated peracids are formed via a HOM formation mechanism and are aware that their formation pathway is different. We hope that the separation in the text where we describe (1) the potential usefulness of peracids as HOMs *proxies* for analytical purposes due to structural similarities and (2) the investigation of peracids as relevant compounds for atmospheric chemistry is clearer now (p.2).

A comparison between experiments and MCM is given in the text at the beginning of 3.3.1. As this is a minor aspect of the paper, we do not show a respective figure.

Page 7, line 22: The yield should be also given as fraction in SOA. This is very common in studies investigating the contribution of peroxides to SOA formation. This would also enable a comparison to other literature studies which is also missing in the manuscript. Based on these values a better discussion of the effect of RH can be done.

As stated several times above we do *not* give absolute quantitative values for the concentrations of the peroxy acid standards in this paper. Therefore, we do not provide yields for these compounds.

Page 7, line 31: A comparison to literature studies is largely missing, e.g. Huang et al., 2013 investigated the influence of RH on hydroperoxides. The authors should also include other potential mechanism/precursor that can be affected by RH, like the Criegee intermediate.

We have added a brief discussion of other investigations of the effect of humidity on peroxy compounds in the text (p.9, line 12).

*"The fact that the peroxypinic acid yield per SOA mass does not depend on humidity agrees with observations made by Docherty et al. (2005), who found no dependence of the organic peroxide yield per SOA mass on humidity. Previous studies of the humidity dependence of individual peroxy compounds were focused on small molecules predominantly residing in the gas phase (e.g. Hasson et al., 2001; Huang et al., 2013) and are therefore not directly comparable with our results. However, the fact that different correlations with humidity were found for different peroxy compounds demonstrates the need for investigation of individual compounds."*

We now also refer to Criegee as potential aspect where humidity could affect the experiments described here (p.8, line 28)

*"This confirms the unimportant role for water vapour in the gas-phase formation of peroxypinic acid not only as a reactant, but also in terms of indirect effects on e.g. the concentrations of precursor species such as Criegee intermediates."*

Page 8, line 15: It is stated that peroxopinic acid degrades over time. Which other products were observed? I would expect the formation of decomposition products like pinic acid. How were the samples treated between the repetitive analysis? Were they stored in a fridge or at room temperature? Were they always protected against UV light?

We agree that formation of pinic acid upon peracid degradation is to be expected and we do observe its occurrence for the synthesized standard (see discussion of NMR results). However, the monoperoxy acid signal is very small in the SOA samples compared to the signal of pinic acid, so that even at full conversion from peracid to acid, the change to the acid signal would be too low to be detected. In general, the relatively minor contribution of monoperoxypinic acid to the total aerosol mass means that assignment of decomposition products was not feasible in this study and is of minor importance.

Filter samples were stored at room temperature between repeat measurements (as stated in line 11, page 4 of the original manuscript) and protected from UV radiation during storage (now mentioned on p.5, line 10):

*"The three additional composite samples were extracted after being stored in their filter boxes at room temperature and under protection from UV radiation for up to 70 h to simulate typical field sampling conditions"*

Minor comments
Page 1, Line 21: change particle phase to particle-phase
Fixed.

Page 2, Line 26: According to IUPAC nomenclature "Sulphuric acid" should be changed to "sulfuric acid". This is also the case in Page 3, line 17.
Fixed.

**Reply to anonymous referee #3**

1. Page 3, lines 23–25: The concentrations of a-pinene and ozone used in these experiments were extremely high. Is it possible that a-pinene partitions to particles, walls, or the filter and that some of the reaction occurs there? Some discussion of the differences between reactions conducted under these conditions and at more typical atmospheric concentrations seems warranted.
Alpha-pinene has a vapour pressure of 633 Pa (US EPA) which corresponds to a vapour saturation concentration ($C_{aP}$) of $3.5 \times 10^7$ μg/m$^3$. Although the aerosol mass loading in the flow tube is high ($C_{OA}$ ~$5 \times 10^4$ μg/m$^3$), equilibrium partitioning theory (e.g. Kroll and Seinfeld, 2008) predicts only a small fraction (<1%) of alpha-pinene should partition into the particle phase under these conditions. However, we agree that condensed material on the flow tube walls and filter may allow even very volatile species to partition due to the large volume of material present (Matsunaga and Ziemann, 2010). Wall reactions of alpha-pinene are unlikely to directly modify the observed aerosol composition since lower volatility products should not repartition from the walls. Condensed phase reactions on the filter may be important, although the charcoal denuder should at that point have removed the majority of O$_3$. The text was amended as follows (p.4, line 23):

*"The average particle mass concentration in the flow tube was about $5 \times 10^4$ μg·m$^3$, assuming a density of 1 g·cm$^3$, with a mode of 200 nm for the number concentration. Under these conditions, α-pinene partitioning to the particles is still negligible (<1%) and while wall partitioning could be significant, it is unlikely to directly modify the observed aerosol composition due to the lower volatility of the products. The produced α-pinene SOA was collected on Durapore® membrane filters (0.1 μm pore size, 47 mm diameter, Merck) for a sampling period of 45 min. Partitioning of α-pinene followed by condensed-phase reactions on the filter might occur and could change the aerosol composition compared to lower mass loadings, although the charcoal denuder should have removed the majority of organic gases and O$_3$, making this less likely."*

2. Page 3, line 26: For these reactant concentrations the ozone should be gone in a few seconds, so the statement that the reaction time is 6.25 min could be clarified. This may give the time for particle-phase reactions, but these will then continue after collection on the filter.
To clarify, the text has been changed as follows (p.4, line 14):
*"Gaseous α-pinene was introduced into the flow tube by passing N$_2$ (200 mL/min) over 500 μL of liquid α-pinene (about 340 ppm initial concentration, measured by PTR-MS according to the procedure described in Giorio et al. (2017)), which results in a residence time of approximately 6.3 min. Under these conditions, the reaction is limited by O$_3$, which according to model calculations is consumed within ~20 s under both humid and dry conditions. The lifetime of O$_3$ and α-pinene in the flow tube was estimated using the AtChem (http://atchem.leeds.ac.uk) numerical box-model alongside the Master Chemical Mechanism (MCM) v3.3.1 (http://mcm.leeds.ac.uk) (Jenkin et al., 1997; Saunders et al., 2003)."*

3. Because of the unstable nature of peroxides it seems that some of the conditions in the HPLC-MS/MS analysis could impact the analysis. For example, the use of 0.1% formic acid, and heater and capillary temperatures of 250 C and 275 C. Please comment on this.
Presence of formic acid should not negatively influence peroxy acid stability. Synthesis of the peroxy acids is carried out under strongly acidic conditions and they are generally more stable at low pH.

While the influence of source temperature was not explicitly tested for peroxypinic acid, we did test it for a wide range of other peroxy acids during a previous study (Steimer et al., 2017). While this is not explicitly mentioned in that paper, we found that peak area improved with increasing temperature, likely due to improved solvent evaporation outweighing any potential decomposition. We did not test the effects of the capillary temperature and can therefore not exclude thermal degradation in the mass spectrometer transfer line.

4. Did the authors consider measuring the total peroxide content of their SOA so that they could estimate the fraction of total peroxides that their molecular analysis detects?
This is a good idea but outside the scope of the present study.

5. Might it be possible to collect particles in a cooled filter apparatus in order to reduce the decomposition of peroxides?
This should be possible and would be a good idea for future experiments. However, in the present study we explicitly decided to perform the collection at room temperature to better simulate the conditions under which samples are usually collected in the field.

**References**

Docherty, K. S., Wu, W., Lim, Y. Bin and Ziemann, P. J.: Contributions of organic peroxides to secondary aerosol formed from reactions of monoterpenes with $O_3$, Environ. Sci. Technol., 39(11), 4049–4059, doi:10.1021/es050228s, 2005.

Eddingsaas, N. C., Loza, C. L., Yee, L. D., Seinfeld, J. H. and Wennberg, P. O.: α-pinene photooxidation under controlled chemical conditions – Part 1: Gas-phase composition in low- and high-$NO_x$ environments, Atmos. Chem. Phys., 12(14), 6489–6504, doi:10.5194/acp-12-6489-2012, 2012.

Giorio, C., Campbell, S. J., Bruschi, M., Tampieri, F., Barbon, A., Toffoletti, A., Tapparo, A., Paijens, C., Wedlake, A. J., Grice, P., Howe, D. J. and Kalberer, M.: Online Quantification of Criegee Intermediates of α-Pinene Ozonolysis by Stabilization with Spin Traps and Proton-Transfer Reaction Mass Spectrometry Detection, J. Am. Chem. Soc., 139(11), 3999–4008, doi:10.1021/jacs.6b10981, 2017.

Hasson, A. S., Orzechowska, G. and Paulson, S. E.: Production of stabilized Criegee intermediates and peroxides in the gas phase ozonolysis of alkenes: 1. Ethene, trans-2-butene, and 2,3-dimethyl-2-butene, J. Geophys. Res. Atmos., 106(D24), 34131–34142, doi:10.1029/2001JD000597, 2001.

Huang, D., Chen, Z. M., Zhao, Y. and Liang, H.: Newly observed peroxides and the water effect on the formation and removal of hydroxyalkyl hydroperoxides in the ozonolysis of isoprene, Atmos. Chem. Phys., 13(11), 5671–5683, doi:10.5194/acp-13-5671-2013, 2013.

Kroll, J. H. and Seinfeld, J. H.: Chemistry of secondary organic aerosol: Formation and evolution of low-volatility organics in the atmosphere, Atmos. Environ., 42(16), 3593–3624, doi:10.1016/j.atmosenv.2008.01.003, 2008.

Matsunaga, A. and Ziemann, P. J.: Gas-wall partitioning of organic compounds in a teflon film chamber and potential effects on reaction product and aerosol yield measurements, Aerosol Sci. Technol., 44(10), 881–892, doi:10.1080/02786826.2010.501044, 2010.

Mentel, T. F., Springer, M., Ehn, M., Kleist, E., Pullinen, I., Kurtén, T., Rissanen, M., Wahner, A. and Wildt, J.: Formation of highly oxidized multifunctional compounds: Autoxidation of peroxy radicals formed in the ozonolysis of alkenes - Deduced from structure-product relationships, Atmos. Chem. Phys., 15(12), 6745–6765, doi:10.5194/acp-15-6745-2015, 2015.

Mutzel, A., Poulain, L., Berndt, T., Iinuma, Y., Rodigast, M., Böge, O., Richters, S., Spindler, G., Sipilä, M., Jokinen, T., Kulmala, M. and Herrmann, H.: Highly Oxidized Multifunctional Organic Compounds Observed in Tropospheric Particles: A Field and Laboratory Study, Environ. Sci. Technol., 49(13), 7754–7761, doi:10.1021/acs.est.5b00885, 2015.

Rissanen, M. P., Kurtén, T., Sipilä, M., Thornton, J. A., Kausiala, O., Garmash, O., Kjaergaard, H. G., Petäjä, T., Worsnop, D. R., Ehn, M. and Kulmala, M.: Effects of chemical complexity on the autoxidation mechanisms of endocyclic alkene ozonolysis products: From methylcyclohexenes toward understanding α-pinene, J. Phys. Chem. A, 119(19), 4633–4650, doi:10.1021/jp510966g, 2015.

Steimer, S. S., Kourtchev, I. and Kalberer, M.: Mass spectrometry characterization of peroxycarboxylic acids as proxies for reactive oxygen species and highly oxygenated molecules in atmospheric aerosols, Anal. Chem., 89(5), 2873–2879, doi:10.1021/acs.analchem.6b04127, 2017.

Szmigielski, R., Surratt, J. D., Vermeylen, R., Szmigielska, K., Kroll, J. H., Ng, N. L., Murphy, S. M., Sorooshian, A., Seinfeld, J. H. and Claeys, M.: Characterization of 2-methylglyceric acid oligomers in secondary organic aerosol formed from the photooxidation of isoprene using trimethylsilylation and gas chromatography/ion trap mass spectrometry, J. Mass Spectrom., 42(1), 101–116, doi:10.1002/jms.1146, 2006.

Witkowski, B. and Gierczak, T.: Analysis of α-acyloxyhydroperoxy aldehydes with electrospray ionization-tandem mass spectrometry (ESI-MSn), J. Mass Spectrom., 48(1), 79–88, doi:10.1002/jms.3130, 2013.

Yasmeen, F., Vermeylen, R., Szmigielski, R., Iinuma, Y., Böge, O., Herrmann, H., Maenhaut, W. and Claeys, M.: Terpenylic acid and related compounds: Precursors for dimers in secondary organic aerosol from the ozonolysis of α-and β-pinene, Atmos. Chem. Phys., 10(19), 9383–9392, doi:10.5194/acp-10-9383-2010, 2010.

Yasmeen, F., Szmigielski, R., Vermeylen, R., Gómez-González, Y., Surratt, J. D., Chan, A. W. H., Seinfeld, J. H., Maenhaut, W. and Claeys, M.: Mass spectrometric characterization of isomeric terpenoic acids from the oxidation of α-pinene, β-pinene, d-limonene, and $\Delta^3$-carene in fine forest aerosol, J. Mass Spectrom., 46(4), 425–442, doi:10.1002/jms.1911, 2011.

Zhao, R., Kenseth, C. M., Huang, Y., Dalleska, N. F. and Seinfeld, J. H.: Iodometry-Assisted Liquid Chromatography Electrospray Ionization Mass Spectrometry for Analysis of Organic Peroxides: An Application to Atmospheric Secondary Organic Aerosol, Environ. Sci. Technol., acs.est.7b04863, doi:10.1021/acs.est.7b04863, 2018.

Zhou, S., Rivera-Rios, J. C., Keutsch, F. N. and Abbatt, J. P. D.: Identification of Organic Hydroperoxides and Peroxy Acids Using Atmospheric Pressure Chemical Ionization – Tandem Mass Spectrometry (APCI-MS/MS): Application to Secondary Organic Aerosol, Atmos. Meas. Tech. Discuss., 2018, 1–17, doi:10.5194/amt-2017-400, 2018.

Ziemann, P. J.: Formation of alkoxyhydroperoxy aldehydes and cyclic peroxyhemiacetals from reactions of cyclic alkenes with $O_3$ in the presence of alcohols, J. Phys. Chem. A, 107(12), 2048–2060, doi:10.1021/jp022114y, 2003.

---

## Referee Report (RR1)

Re-review of ms acp-2017-1197

The authors have satisfactorily addressed the concerns of this reviewer adequately and publication is recommended. A few minor comments described below remain to be addressed.

p. 5, line 10. The extraction procedure appears to have been described in the previous page, line 31?

p. 6, line 21. artefacts → artifacts.

p. 6, line 31 – p. 7 lines 1 and 2. Unless the authors are referring here to the relative yields of the monoperoxypinic acid II under dry and wet conditions, this procedure does not seem to be valid for quantitation of isomer II, but rather a measure of consistency in instrument performance and sample preparation using camphoric acid as an internal standard. Quantitation relies on the assumption of (1) the same response factor for the base peaks of the MS/MS spectra of camphoric acid and isomer II of monoperoxypinic acid and (2) the base peaks of the MS/MS spectra of both camphoric acid (loss of $CO_2$) and monoperoxypinic acid isomer II (loss of $H_2O$) represent the same branching ratios of leading to the monitored product ions. If relative peak areas (yields) of isomer II under different conditions or in different experiments are being compared, then this approach is valid. The authors need to clarify this and not refer to the quantitation of isomer II.

p. 7, line 7. Figure 3 is described in the caption as the "base peak chromatogram" of the synthetic mixture. The authors might supply the specific *m/z* of the base peak.

p. 8, lines 18 – 19. Are integrals being compared derived from the molecular ions in full scan MS, or from base peaks of MS/MS spectra? It would seem that the authors can only estimate relative yields of monoperoxypinic acids. Given this stipulation, the conclusion that RH makes no difference is valid, since selected specific ions can be directly compared between dry and wet conditions.

---

## Author Response (AR2)

**Reply to referee report 1 (Re-review of ms acp-2017-1197)**

The authors have satisfactorily addressed the concerns of this reviewer adequately and publication is recommended. A few minor comments described below remain to be addressed.

p. 5, line 10. The extraction procedure appears to have been described in the previous page, line 31?

That is correct. The text has therefore been changed to (p.5, line 10): *"The first composite sample was extracted immediately, following the same procedure as previously described."*

p. 6, line 21. artefacts → artifacts.

To my best knowledge, this is just a difference in British vs. American spelling. As the rest if the manuscript also uses British English, artefacts should be the correct spelling.

p. 6, line 31 – p. 7 lines 1 and 2. Unless the authors are referring here to the relative yields of the monoperoxypinic acid II under dry and wet conditions, this procedure does not seem to be valid for quantitation of isomer II, but rather a measure of consistency in instrument performance and sample preparation using camphoric acid as an internal standard. Quantitation relies on the assumption of (1) the same response factor for the base peaks of the MS/MS spectra of camphoric acid and isomer II of monoperoxypinic acid and (2) the base peaks of the MS/MS spectra of both camphoric acid (loss of $CO_2$) and monoperoxypinic acid isomer II (loss of $H_2O$) represent the same branching ratios of leading to the monitored product ions. If relative peak areas (yields) of isomer II under different conditions or in different experiments are being compared, then this approach is valid. The authors need to clarify this and not refer to the quantitation of isomer II.

We are indeed comparing relative yields under varying conditions. We agree that quantitation can be misleading in this context and have therefore reworded the sentence as follows (p.6, line 31, p.7, line 1): *"The integrated peak of the extracted ion chromatogram of m/z 183.06 was used to determine the relative yield of monoperoxypinic acid isomer II."*

p. 7, line 7. Figure 3 is described in the caption as the "base peak chromatogram" of the synthetic mixture. The authors might supply the specific *m/z* of the base peak.

The base peak chromatogram is from the MS measurement and based on the full mass range, not a specific m/z. To clarify this, the caption of Figure 3 (p.17) has been modified as follows: *"Figure 3: Base peak chromatogram of the synthesised mixture for mass range m/z 100–650, showing the separation of pinic acid (1), monoperoxypinic acid isomers I and II (2, 3) and diperoxypinic acid (4)."*

p. 8, lines 18 – 19. Are integrals being compared derived from the molecular ions in full scan MS, or from base peaks of MS/MS spectra? It would seem that the authors can only estimate relative yields of monoperoxypinic acids. Given this stipulation, the conclusion that RH makes no difference is valid, since selected specific ions can be directly compared between dry and wet conditions.

As described at the bottom of page 6, we are using the integral of the peak in the extracted ion chromatogram of *m/z 183.06* (the main fragment of monoperoxypinic acid isomer II) to determine the relative yields. While we have already stated that we are comparing relative yields (p. 8, line 21 of the revised markup version of the manuscript), we have now made sure to use this term consistently (p. 7, line 1 and p. 9, line 6) to avoid confusion about the procedure.

[revised manuscript text omitted]